



# Seasonal Predictability of Vapor Pressure Deficit in the western United States

Melissa L. Breeden[1], Andrew Hoell[1], Rochelle P. Worsnop[1], John R. Albers[1], Michael T. Hobbins[1,2], Rachel M. Robinson[1,2], and Daniel J. Vimont[3],

[1]NOAA Physical Sciences Laboratory, Boulder, 80305, USA
[2]Cooperative Institute for Research in Environmental Sciences, University of Colorado Boulder, Boulder, 80309, USA
[3]Department of Atmospheric and Oceanic Sciences, University of Wisconsin-Madison, Madison, 53706, USA

*Correspondence to*: Melissa L. Breeden (melissa.breeden@noaa.gov)

**Abstract.** Saturation vapor pressure deficit (VPD), a measure of the difference between how much moisture the atmosphere can hold versus how much is present, is highly correlated with the annual mean area burned by wildland fires in the western United States. The present analysis uses linear inverse models (LIMs) to forecast seasonal VPD and decompose skill into contributions from a nonlinear trend, coupled sea surface temperature (SST)-VPD variability, and VPD-only variability.
Subregions of the western US are considered using Geographic Area Coordination Centers which have different times of year and lead times for which VPD forecast skill is greatest. However, the sources of skill are similar among the subregions. In LIM forecasts, particularly those made for summer and early fall, the trend contributes to VPD skill up to 18 months in advance, with a secondary contribution from internal VPD variability at lead times of one to two months. Positive SST and VPD anomalies and negative soil moisture anomalies are associated with the positive sign of the trend time series, which has
been observed without interruption since the late 1990s. Coupled SST-VPD variability contributes to VPD skill mainly for forecasts verifying between December to May for lead times up to 12 months in some subregions. Forecasts that are especially skillful and display high confidence, seasonal forecasts of opportunity (SFOs), are associated with SSTs that produce high VPD skill over California, the Southwest, and Texas, while internal VPD anomalies contribute to skill over the Great Basin and western Northern Plains. SFOs are initialized during periods of El Niño-Southern Oscillation development,
with La Niña SSTs associated with positive western US VPD anomalies and consequently, enhanced wildland fire risk.

## 1 Introduction

In the continental United States, wildland fires have increased in terms of frequency and area burned in recent years (Westerling et al. 2006; Higuera and Abatzoglou 2021; NIFC, 2022), with the "Western Wildfires of 2021" landing on the U.S. 2021 and 2022 Billion-Dollar Weather and Climate Disasters list (NOAA NCEI, 2021-2022). The immediate physical
danger and damage from wildfire impacts on air quality and the water supply pose serious threats to public health and well-being (Fifth National Climate Assessment Chapters 25, 27, 28; Williams et al. 2022) and to sensitive ecosystems (Coop et al. 2020; Yuan et al. 2019). Past studies have revealed a positive and strong relationship between water vapor pressure deficit (VPD) to annual-mean burned area in the western US (Williams et al. 2014; Abatzoglou and Williams, 2016; Higuera and



Abatzoglou 2021; Zhuang et al. 2021; Buch et al 2023), while higher nighttime VPD values have been linked to longer fire
days in recent decades (Chiodi et al. 2021). VPD (1) is the difference between saturation vapor pressure ($e_s$) and actual
vapor pressure ($e_a$), and is therefore a function of both temperature and humidity. It has long been recognized as an absolute
measure of the near-surface atmospheric moisture difference (Anderson 1936) with larger differences associated with
increased fuel aridity and higher burned area.

Despite VPD's relation to the area burned by wildland fires, sources of seasonal (> one month lead time) VPD predictability
have not been heavily studied, and it is unclear, given how the two components of VPD are related to different climate
patterns, what will influence the predictability of the combination of temperature and moisture that VPD represents. Through
the large-scale and persistent circulation anomalies it can produce, the El Niño-Southern Oscillation (ENSO) has been linked
to western US VPD anomalies and drought; anomalous anticyclonic flow and subsidence associated with La Niña conditions
warms and dries the air column ultimately enhancing southwest US VPD (Hoell et al. 2014; Seager et al. 2015; Mankin et al.
2021; Seager et al. 2022; Hoell et al. 2022). Many studies have highlighted ENSO's seasonal predictability and its relevance
for precipitation and temperature forecasts (e.g., Barnett and Priesendorfer 1987; Barnston 1994; Higgins et al. 2004; Quan
et al. 2006; Tian et al. 2014; Newman and Sardeshmukh 2017; L'Heureux et al. 2019; L'Heureux et al. 2020; Becker et al.
2022), although these links can depend on the time period studied and index used to capture ENSO (Wang and Kumar 2015;
Patricola et al. 2020). The long-term temperature trend enhances VPD by increasing $e_s$ and influences the skill of
temperature-related variables (Peng et al. 2013; Risbey et al. 2021). Conversely, $e_a$ does not appear affected by the upward
temperature trend and was in fact found to decrease from 1981-2020 (Simpson et al. 2023). Finally, soil moisture, a slowly-
evolving quantity that is the aggregate effect of precipitation, vegetation, and surface fluxes on the moisture content of the
soil, may also be a source of VPD predictability given its high temporal autocorrelation at depth (Rahman et al. 2015) and
the potential 'reemergence' of soil moisture anomalies months after they first develop (Kumar et al. 2019).

We use an empirical dynamical model – namely a linear inverse model (LIM; Penland and Sardeshmukh 1995) – to test the
hypothesis that sources of VPD predictability include the long-term warming trend and ENSO, given their relationship to
VPD variability. A LIM decomposes climate anomalies into a set of nonorthogonal eigenmodes, or 'empirical normal
modes' (Hasselmann 1988; hereafter referred to as 'modes' or 'LIM modes'). Due to their nonorthogonality, these structures
can evolve via interference to produce patterns of rapid anomaly growth or decay (Penland and Sardeshmukh 1995; Farrell
and Ioannou 1996; Henderson et al. 2020; see Albers et al. 2022 for a recent discussion). LIM modes can be used to
disentangle how specific processes contribute to predictability on seasonal and subseasonal timescales (Penland and
Matrosova 1994; Penland and Matrosova 2006; Sardeshmukh et al. 2000; Newman et al. 2011, Henderson et al. 2020,
Albers and Newman 2021; Breeden et al. 2022a). Seasonal SST and ENSO forecasts produced with a LIM can be as skillful
as those from operational forecast models (Newman and Sardeshmukh 2017), suggesting LIM may be useful for seasonal
VPD prediction. Since LIM forecasts here are built from reanalysis inputs, they do not have biases that exist in seasonal





forecast models such as those related to ENSO (Beverley et al. 2022). This is advantageous for a variable such as VPD, a quantity that climate models struggle to simulate, given mean biases in the representation of $e_a$ (Simpson et al. 2023) and

that operational seasonal forecast models have minimal low-level humidity skill (McEvoy et al. 2015). Given the aforementioned skill of LIM-based SST forecasts and the link between SST and VPD in this region (e.g., Seager et al. 2015), we use a LIM to represent and forecast seasonal VPD.

This article is organized as follows. Methods outlining the reanalysis data used in this study and the processes to construct

LIM forecasts are discussed in Section 2. Section 3.1 examines seasonal western US VPD forecast skill and contributions from a nonlinear trend, Section 3.2 considers detrended forecast skill and contributions from coupled SST-VPD and VPD-only variability, and Section 3.3 considers Seasonal Forecasts of Opportunity (SFOs) and their relationship to ENSO.

## 2 Data and Methods

### 2.1 Data

We use monthly mean Japanese Meteorological Agency 55-year Reanalysis (JRA-55; Kobayashi et al., 2015) SST, top three levels of soil moisture (SM), 2-meter air temperature (2mT), and 2-meter relative humidity (RH) from 1958 – 2021. 2mT and RH are used to calculate saturation vapor pressure $e_s$, actual vapor pressure $e_a$, and subsequently VPD (Eq. 1-2):

$$VPD = e_s(T) - e_a \qquad (1)$$

$$e_a = e_s * \left(\frac{RH}{100}\right) \qquad (2).$$

The horizontal resolution and domains used for each variable are shown in Tables 1-2. VPD, SST, and SM anomalies are calculated with respect to the 1958-2021 monthly means; a three-month running average is applied to the anomalies thereafter.

VPD skill averaged over the western US is assessed using Geographic Area Coordination Centers (GACCs), or subregions,

that are used in resource planning and emergency management by the National Interagency Fire Center (Fig. 1; NIFC; https://gacc.nifc.gov/). This analysis considers skill averaged over the Northern and Southern California, Great Basin, Southwest, and Northwest GACCs, as well as over each of these subregions separately.





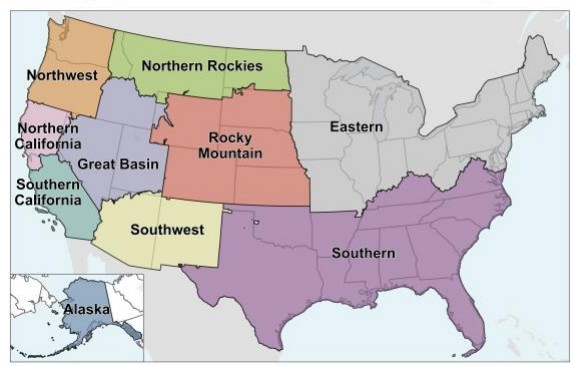

**Figure 1: Geographic Area Coordination Centers used by the National Interagency Coordination Center.**


## 2.2 Linear Inverse Model

This study uses two Linear Inverse Models (LIMs): one trained on anomalies that include a trend and one using detrended anomalies. For each LIM, the LIM operator and LIM-based retrospective forecasts are calculated using the same procedure (Sections 2.2.1-2.2.2). The dynamical filtering approach (Section 2.2.3) is used several ways: to identify a nonlinear trend

and evaluate trend-related skill, detrend anomalies for a second LIM, and decompose detrended VPD forecast skill. Seasonal forecasts of opportunity are calculated for only the detrended LIM as described in Section 2.2.4.

### 2.2.1 LIM Operator

A LIM models the evolution of a subset of climate anomalies defined by the state vector $x$ (Eq. 3) using a linear approximation to the (fully nonlinear) slowly evolving dynamics ($Lx$) and rapidly decorrelating stochastic processes $F_s$ (Eq.

4-5; Penland 1989; Penland and Sardeshmukh 1995). The linear approximation of the predictable dynamics is determined by (Eq. 5) using the lagged $C_{\tau_o}$ and zero-lag $C_0$ covariance between the subset of relevant system variables represented by $x$. A three-month lag $\tau_o$ is chosen to calculate $C_{\tau_o}$ and subsequently $L$ (e.g., Alexander et al. 2008; Newman et al. 2011):

$$x = \{SST, SM, VPD\} \qquad (3)$$

$$\frac{dx}{dt} = Lx + F_s \qquad (4)$$

$$L = ln(C_{\tau_o} * inv(C_0))/\tau_o \qquad (5)$$

To reduce the dimensionality of $x$ and consequently $L$, each variable is considered in terms of empirical orthogonal functions (EOFs) whose expansion coefficients, or principal components (PCs), are used to construct $x$, $L$, and the EOF-truncated



forecasts. Enough PCs are included to retain the majority of each variable's variance over its respective domain (Tables 1-2).

Several tests are required to ensure each LIM is numerically stable and exhibits the properties of the linear approximation to

the system; these tests, including the tau-test, are detailed in Penland and Sardeshmukh (1995), and are conducted for both

LIMs used here to confirm the suitability of the LIM to model the variables in $x$ (Supplement Section a; Figs. S1-S2, Table

S1).

| Variable | Domain | Horizontal Resolution | # EOFs retained (% variance explained) |
|----------|--------|----------------------|---------------------------------------|
| SST | 55°S – 55°N, 0-358.75°E | 1.25°x1.25° | 8 (54%) |
| SM | 24°N-50°N, 230-300°E | 2°x2° | 6 (61%) |
| VPD | 25°N-50°N, 230-300°E | 0.5°x0.5° | 9 (78%) |

**Table 1: Description of variables used to construct the VPD LIM that includes the trend. JRA-55 Reanalysis is used**
**for each variable.**

| Variable | Domain | Horizontal Resolution | # EOFs retained (% variance explained) |
|----------|--------|----------------------|---------------------------------------|
| SST | 25°S – 55°N | 1.25°x1.25° | 13 (69%) |
| VPD | 25°N-50°N, 230-300°E | 0.5°x0.5° | 12 (82%) |

**Table 2: Description of variables used to construct the *detrended* VPD LIM. JRA-55 Reanalysis is used for each**
**variable. Note that it is the variance of the detrended anomalies that is indicated.**

### 2.2.2 VPD Forecasts

Retrospective forecasts $\hat{x}(\tau)$ of $x$ are generated for a given lead time $\tau$, which ranges from 1-18 months, by integrating the

homogeneous component of (Eq. 4) using the forecast operator $G(\tau)$ and initial conditions $x(0)$:

$$\hat{x}(\tau) = x(0)exp(L\tau) = x(0)G(\tau) \qquad (6).$$

Cross-validated forecasts are generated by dividing the data into 16 'folds,' removing a single fold (4 years) at a time,

recalculating $L$ and using it to generate forecasts for the removed four-year period (e.g., Breeden et al. 2022a,b; Albers and

Newman 2019). To assess LIM forecast skill, we compute the anomaly correlation coefficient (ACC) between time series of

the EOF-truncated forecast VPD and full-field (i.e., 100% variance) observed VPD at each grid point. The two LIMs

calculated using anomalies with or without the trend included were used to generate separate forecasts. The LIM forecasts

containing the trend are verified against anomalies also containing the trend, while the detrended LIM forecasts are verified

against detrended anomalies. LIM forecasts are also compared to persistence forecasts that are generated using the full-field





VPD anomalies. A persistence forecast assumes there is memory in the system, which may arise from a warming trend, or, say, slowly evolving SSTs that could lead to forecast skill if current, initial conditions (ICs) are well observed. Though they are simple, persistence forecasts remain a competitive baseline that seasonal forecast models struggle to beat for 2mT even

one season in advance (Zhang et al. 2019).

Confidence bounds for the ACC of each set of forecasts are determined nonparametrically using bootstrapping with replacement 10000 times. This is done by resampling the forecasts to create an equally plausible forecast set and calculating the ACC of that set. This process is repeated 10000 times, and the 95% confidence bounds of the resultant ACC estimate are

calculated. When the ACC confidence bounds are greater than the null hypothesis ACC, ACC is denoted with black stippling.

### 2.2.3 Dynamical Filter

To diagnose sources of seasonal VPD skill, we use the dynamical filtering technique described in Penland and Matrosova (1994). This perspective considers climate anomalies to be the manifestation of the interference between the modes of the

system represented by $L$. While the patterns of various modes may project onto one another – in contrast to EOF patterns requiring their orthogonality – their timescales and periods differ, so that the relative contributions from various modes change with time as each mode uniquely evolves. LIM modes occur as either a pair with a decay timescale and period, or as a single, non-oscillatory mode that purely decays.

As in past studies (Penland and Matrosova 1994; Alexander et al. 2008; Seager et al. 2023), we isolate a long-term trend using the eigenmode with the longest e-folding timescale, namely the least-damped eigenmode, with a slight modification, to calculate a time series representing a nonlinear trend (Supplement Section b). The pattern associated with this modified time series is determined by regressing the time series onto the *untruncated* VPD, SST and SM anomalies, respectively, and the combination of the time series and pattern are used to calculate trend-related anomalies for each timestep and variable. We

also consider the trend in the EOF-truncated space of the LIM to filter $x$, which provides the ICs for the forecasts, to contain only the trend mode, $x_{tr}$. Thereafter, trend-filtered forecasts are generated, $\hat{x}_{tr}$, whose skill is evaluated against the unfiltered verification. Finally, we detrend anomalies for a new LIM by subtracting the trend-associated VPD and SST anomalies from the unfiltered anomalies at each gridpoint, which is confirmed to be effective (Figs. S3-S4; Eq. 7).

$$VPD_d = VPD - VPD_{tr} \qquad (7)$$

A similar process is employed to detrend the SST anomalies; thereafter EOFs and PCs are recomputed for both variables to create a detrended state vector $x_d$ and a detrended LIM (Table 2). Detrended SM anomalies were not included in the detrended LIM because they did not improve VPD skill (not shown). Forecasts are generated using the detrended LIM to compare the seasonality and amplitude of seasonal VPD forecast skill with and without the trend included.



To further decompose sources of VPD predictability, the detrended LIM is filtered into its eigenmodes and the relative
amplitude of SST and VPD in each mode assessed (Fig. S5; Newman et al. 2009; Henderson et al. 2020). The most rapidly
decaying eigenmodes are dominated by large (relative) amplitude in VPD, while SST amplitude is low until mode #11 and
increases with increasing e-folding timescale (Table S2). We therefore define two subspaces within the detrended LIM: a
VPD-only subspace composed of the most rapidly decaying modes, modes #1-10, and a coupled SST-VPD subspace

composed of the remaining modes, #11-25. These subspaces are used to filter $\boldsymbol{x}_d$:

$$\boldsymbol{x}_d = \boldsymbol{x}_{d\_VPD_{only}} + \boldsymbol{x}_{d\_SST-VPD} \qquad (8),$$

thereby generating filtered ICs and forecasts associated with the SST-VPD and VPD-only components of $\boldsymbol{x}_d$. The
verification in both cases is the full-field, detrended VPD anomaly field. Note that while the components of $\boldsymbol{x}_d$ are linearly
additive, ACC is a quadratic field and therefore is not. However, the relative importance of each component can still be

evaluated.

### 2.2.4 Seasonal Forecasts of Opportunity

In addition to considering the deterministic skill of all forecasts, the LIM can identify particularly confident and skillful
forecasts, defined as 'seasonal forecasts of opportunity' (SFOs). Each forecast's signal covariance $\boldsymbol{F}(\tau, t)$ and error
covariance $\boldsymbol{E}(\tau)$ are used to calculate the signal-to-noise ratio $S^2$, which is used to anticipate particularly skillful VPD

forecasts at the time the forecast is made. As $S^2$ increases, so does the expected skill $\rho_\infty$ of the forecast (Sardeshmukh et al.
2000; Newman et al. 2003, Albers and Newman 2019; Breeden et al. 2022a,b), as follows:

$$\boldsymbol{F}(\tau, t) = <\hat{\boldsymbol{x}}_d(t + \tau)\hat{\boldsymbol{x}}_d(t + \tau)'> \qquad (9)$$

$$\boldsymbol{E}(\tau) = \boldsymbol{C}_0 - \boldsymbol{G}(\tau)\boldsymbol{C}_0\boldsymbol{G}(\tau)' \qquad (10)$$

$$S^2(\tau, t) = \frac{tr[\boldsymbol{F}(\tau)]}{tr[\boldsymbol{E}(\tau)]} \qquad (11)$$

$$\rho_\infty(\tau, t) = \frac{S^2(\tau)}{\{[S^2(\tau)+1]S^2(\tau)\}^{.5}} \qquad (12),$$

where $\boldsymbol{F}(\tau, t)$ is the forecast signal covariance that varies with lead time $\tau$ and initialization time, t, and $\boldsymbol{E}(\tau)$ is the forecast
error covariance, which is a function of $\tau$ only. In this study, we test if the detrended LIM can identify forecasts of
opportunity for an area averaged over the combined Southern California and Southwest subregions. SFOs are defined as the
forecasts initialized with the top 15% of $S^2$ and therefore $\rho_\infty$. The skill of that subset of forecasts is compared with the

remaining 85% of forecasts, representing 'non-SFOs.' Results are similar for SFOs defined using the top 10-30%. The
detrended LIM filter is used to filter the SFOs for their skill contributions from the SST-VPD and VPD-only subspaces of
the LIM.



Finally, to consider the patterns associated with SFOs, we assess the resemblance of the ICs observed during SFOs to the
'optimal initial conditions' (OPT-ICs; Farrell 1988; Penland and Sardeshmukh 1995; Tziperman and Ioannou 2002) of the
system. OPT-ICS identify the VPD and SST patterns that maximize system growth over a specified period of time. Our
hypothesis is that SFOs are associated with periods when the earth system is conducive to the 'optimal' growth of familiar
patterns such as ENSO (e.g., Penland and Sardeshmukh 1995; Breeden et al. 2022a). OPT-ICs are determined from solving
the eigenvalue problem for a specified time interval or growth period $\tau$,

$$[\boldsymbol{G}(\tau)^T \boldsymbol{G}(\tau)]\boldsymbol{v}(\tau) = \mu(\tau)\boldsymbol{v}(\tau) \qquad (13),$$

where $\boldsymbol{v}(\tau)$ and $\mu(\tau)$ are the eigenvectors and eigenvalues, respectively, representing the patterns and amplification of the
domain-integrated variance of $\boldsymbol{x}_d$. Two patterns associated with the greatest and second-greatest eigenvalues of the system,
'OP1' and 'OP2,' are considered for growth periods of seven and three months, respectively. These growth periods represent
the $\tau$ with the greatest growth (i.e., the largest positive $\mu(\tau)$ of all $\tau$; Fig. S9). To test the hypothesis that SFOs are more
strongly associated with optimal patterns than non-SFOs, the amplitude of the projection of ICs onto OPT-ICs associated
with OP1 and OP2 is compared during SFOs and non-SFOs.

## 3 Results

### 3.1 VPD LIM

VPD skill produced by the LIM that includes the trend peaks in the western US and Mexico during summer and exceeds the
skill of persistence for lead times greater than two months (Fig. 2). Some VPD skill from the LIM is also observed over the
eastern United States, while a skill minimum is located in the Great Plains and upper Midwest (Fig. 2a). ACC is similar
between lead times of three – eighteen months and peaks late in the warm season over the western US (Fig. 2b). High skill
(> 0.5 ACC) at lead times of up to 14 months in a persistence forecast is also observed during MJJ, JJA, and JAS seasons,
but not for all lead times as in the LIM (Fig. 2c). The LIM outperforms persistence at lead times greater than three months
for all verification months, which is notable since the LIM is only trained on 78% of VPD variance while persistence uses
full-field (100% variance) VPD anomalies; both are verified against full-field anomalies. The LIM outperforms persistence
most notably at lead times of around seven and nineteen months, corresponding to forecasts for summer made the prior
winter.





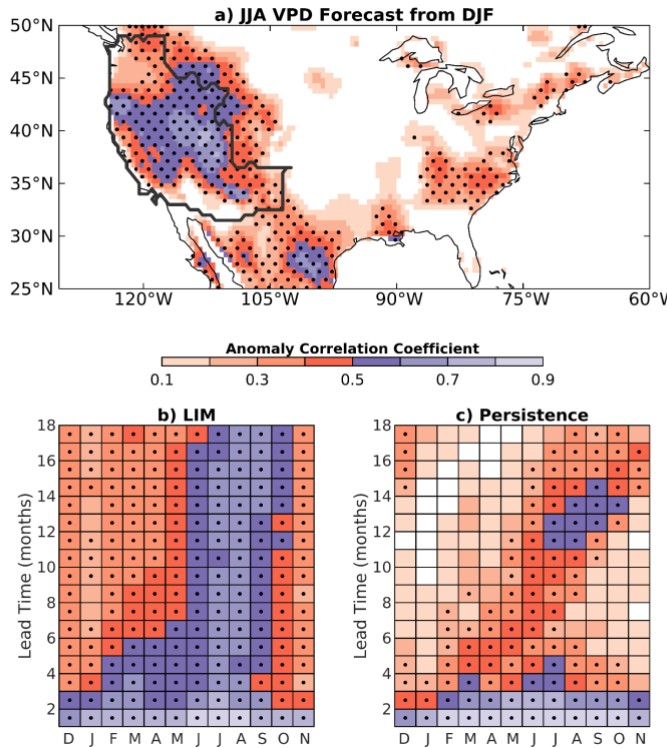

**Figure 2: VPD forecast skill (ACC; anomaly correlation coefficient) of the LIM and persistence forecasts. Panel a) shows LIM forecast skill verifying in Jun-Jul-Aug (JJA) for a six-month lead time, i.e., for DJF initializations. The black line shows the boundary of the region used for area-averaging forecast skill, which includes the Southern and Northern California, Great Basin, Southwest, and Northwest subregions (Fig. 1). Area-averaged ACC over the region covered by these subregions is shown from b) the LIM and c) a full-field (i.e., 100% variance) persistence forecasts. In panels b) and c) the center month of the 3-month verification period is shown on the x-axis, and lead time is on the y-axis. Skill that is different from zero with 95% confidence is indicated by black dots.**

Considering VPD ACC for each western US subregion shows that peak skill may differ in magnitude but occurs during the June-September timeframe everywhere except for the Southwest (Fig 3). Skill is greatest in the Great Basin during JJA-JAS seasons when ACC values exceed 0.6 up to 18 months in advance (Fig. 3e), while skill is lowest in the Southern California region, where ACC rarely exceeds 0.5 (Fig. 3c). The Southwest is unique in that ACC peaks earlier in the year, during boreal spring, at all lead times, with lower skill during boreal summer when the other western US subregions have greatest skill. It is possible that Southwest skill is lower due to the influence of the North American Monsoon, which could reduce VPD predictability in the LIM during summer when the monsoon is active (Prein et al. 2022).



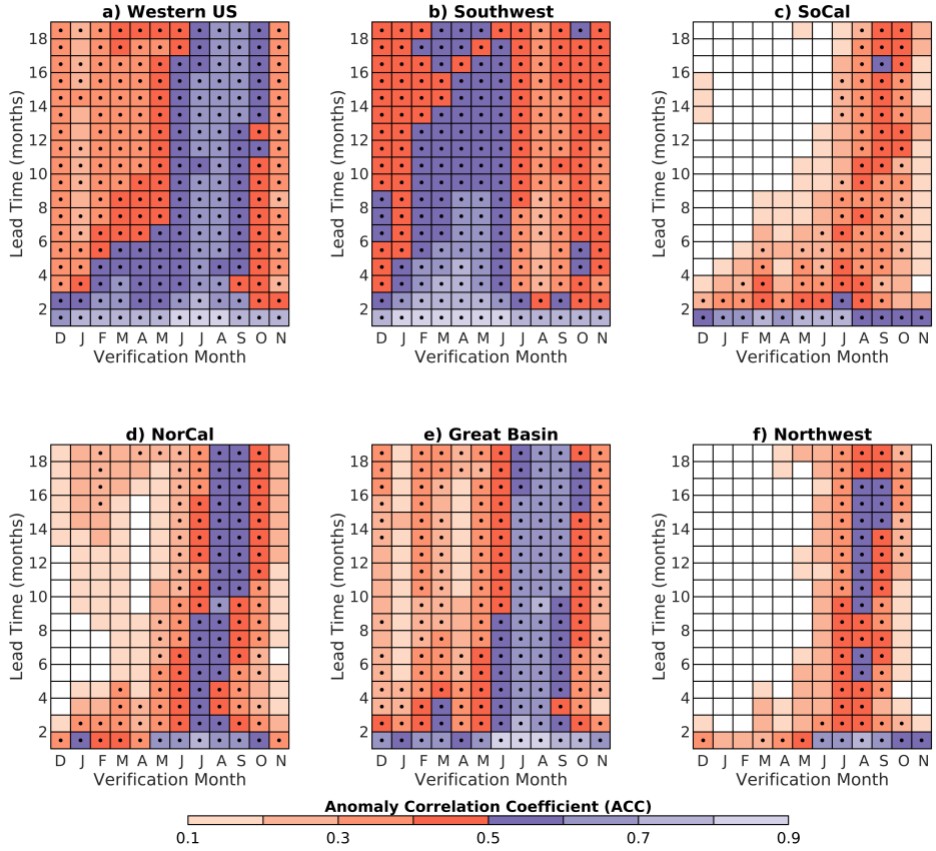

**Figure 3: ACC averaged over a) the western US (Fig. 2a), b) Southwest, c) Southern California (SoCal), d) Northern California (NorCal), e) Great Basin, and f) Northwest subregions for LIM forecasts that included the trend in its suite of unfiltered ICs.**

Given the strong relationship between VPD and 2mT, it is likely that LIM VPD skill is influenced by the long-term warming trend as observed in other forecast systems (Risbey et al. 2021). We use the LIM's modified least damped eigenmode to construct the trend time series which over the 1958-2021 period is not linear (Fig. 4; see supplement for details). Regressing full-field anomalies onto the time series indicates that positive SST and VPD anomalies and negative SM anomalies are associated with the positive sign of the trend, which is observed without interruption after the late 1990's.



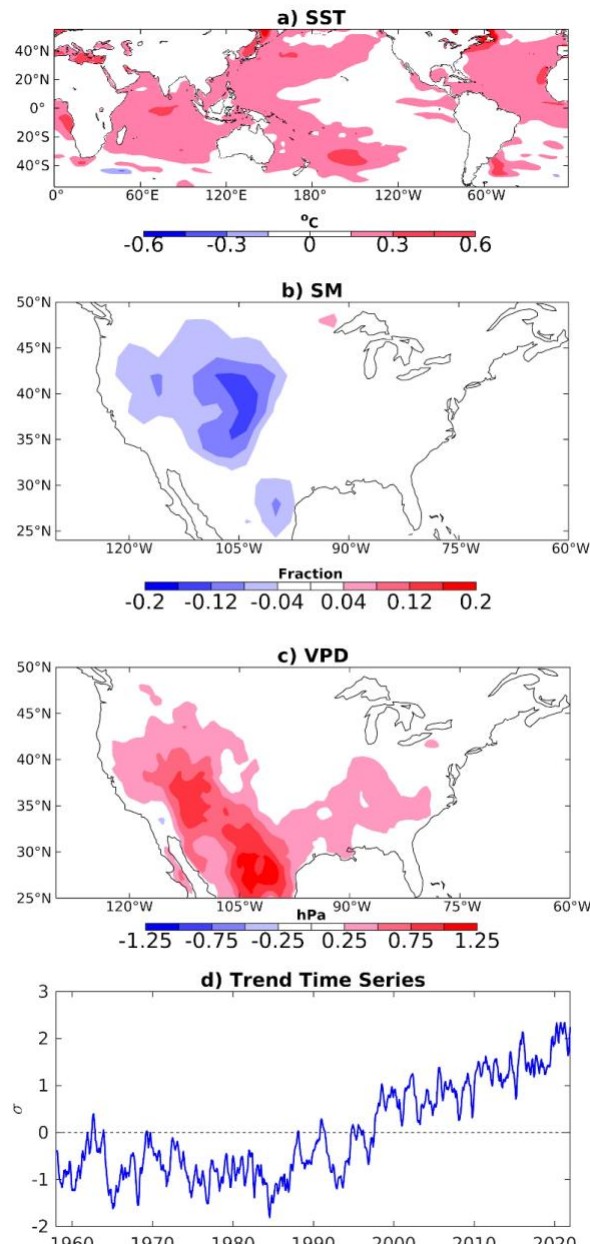

**Figure 4: Characteristics of the trend. Panels a) – c) show the regression patterns and d) time series used to construct an estimate of the trend for a) SST (units degrees C), b) SM (units relative fraction) and c) VPD (units hPa) associated with a +1σ value of the modified trend eigenmode time series shown in panel (d); see supplement for more information.**

Evaluating the skill of the trend mode's contribution to the VPD forecasts confirms our hypothesis that most VPD skill at long lead times is associated with a trend during the warm season (Fig. 5). While both persistence and the LIM contain



information about the trend in their forecasts, the LIM likely outperforms persistence because VPD evolves seasonally. As such, a persistence forecast from winter into summer shows poor skill, as the physical processes that produce winter VPD anomalies are not relevant for the summer VPD forecast. Indeed, winter persistence forecasts display low skill for summer verifications (Fig 2c), while LIM forecasts remain skillful. In contrast to persistence, the LIM captures the *time evolution* of all modes, and since the trend mode decays the most slowly by definition (Table S2), it emerges as the leading pattern in the forecast at longer lead times, which produces high VPD forecast skill in the warm season.

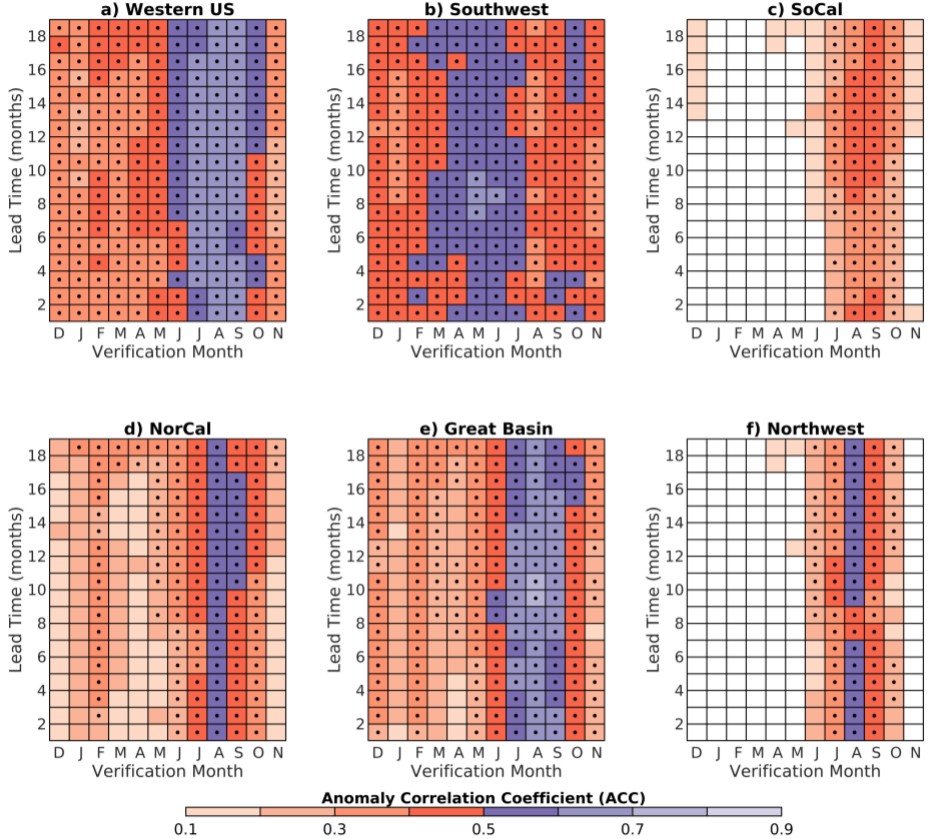

**Figure 5: as in Fig. 3 but for LIM forecasts that *only* include the trend mode in its ICs.**

### 3.2 Detrended LIM

To assess VPD skill beyond the trend, a second LIM is trained using detrended VPD and SST anomalies (see Methods for details). The amplitude and timing of greatest detrended VPD skill differs notably from the LIM including the trend, underscoring the trend's strong influence on skill and VPD in the warm season. Forecasts made for the NDJ – MAM seasons have the highest skill with a minimum observed during JAS-SON, when ACC is negligible at lead times longer than two months over many subregions (Fig. 6). The greatest skill in any season is observed in the Southwest, which has ACC different from zero during the OND – AMJ seasons for lead times up to 17 months but minimal skill in summer and early





fall (Fig. 6b). Conversely, the Northern California subregion only has skill during the AMJ – JJA seasons up to six months in advance (Fig. 6d). Southern California and the Great Basin display skill in both the warm and cool seasons to varying

degrees, while the Northwest subregion has minimal skill beyond a one-month lead time except in the meteorological winter, when VPD anomalies are small (Fig. 6c,e,f). Comparing VPD skill that arises from a LIM trained on *linearly* detrended anomalies, which would be simpler and has been previously done (Vimont 2012, Newman 2013), shows a similar skill pattern but lower skill during the cold season than the LIM trained on anomalies detrended using the LIM's least-damped eigenmode (Fig. S6). This could be because the linear trend is a mixture of the externally forced signal and multidecadal

variability, the latter of which may enhance VPD predictability. As such, linear detrending may convolve the trend with variability.

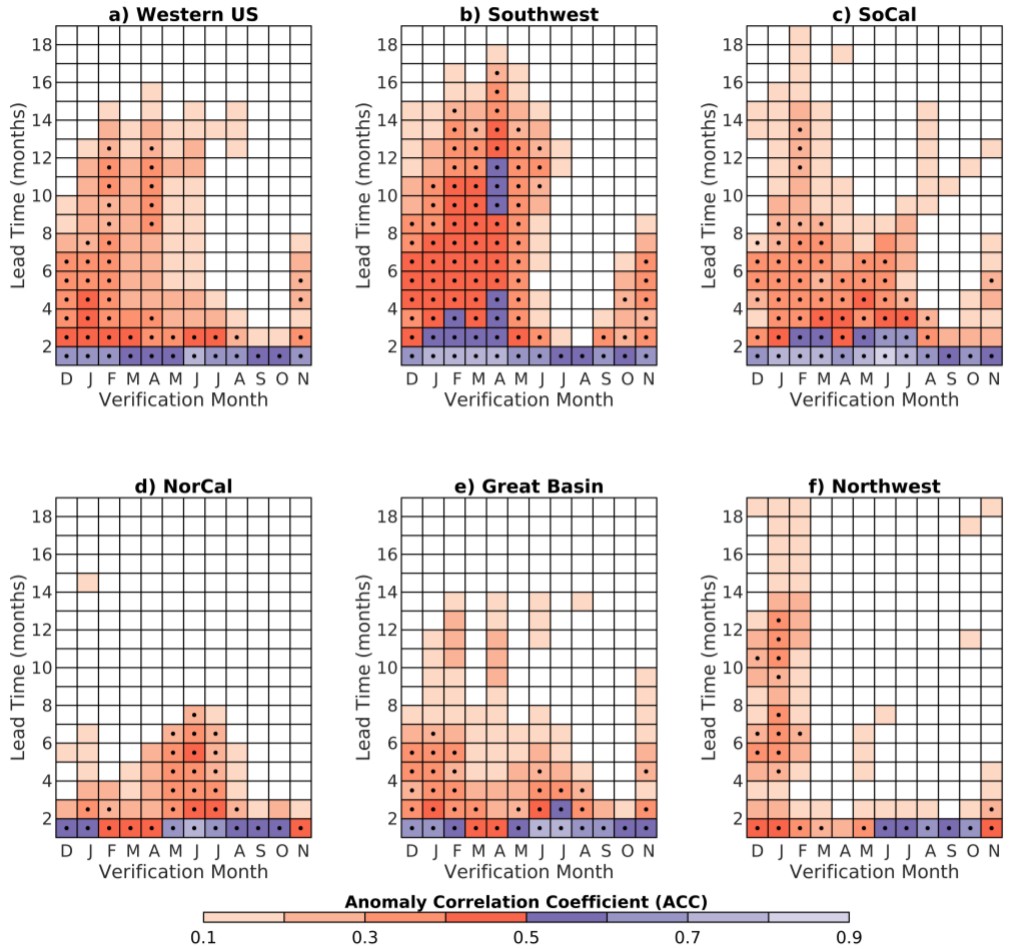

**Figure 6: As in Fig. 3 but for LIM forecasts from the detrended LIM and using unfiltered ICs.**






Despite regional differences in the timing and magnitude, coupled SST-VPD modes are the leading source of detrended VPD skill in all subregions except the Great Basin (Figs. 7-9). We determine which modes of variability contribute most to VPD skill by filtering LIM ICs using the LIM eigenmodes (Section 2.2.3). SST variance resides purely in the coupled SST-VPD modes by definition, while VPD variance is shared almost equally between the two groups (Fig. 7). Total VPD variance does not equal the sum of the variances of the two groups due to the nonnormality of the system (Farrell and Ioannou 1996; Albers and Newman 2021), but their patterns are similar to total variance suggesting that the peaks in VPD variance observed over the southern and western US – particularly Texas and southern California – can be understood as locations with strong interference between VPD-only and SST-VPD modes. VPD skill is associated mainly with SSTs, with a secondary contribution from VPD-only modes at shorter lead times (c.f., Fig. 6, Fig. 8, Fig. 9). The Great Basin and Southwest have the greatest contributions from the VPD-only modes, though there are only rare occurrences of skill greater than zero observed beyond a one-month lead time. SST-VPD modes have slower decay rates (Table S2) given the high autocorrelation of SSTs, making it unsurprising that these contribute more to seasonal VPD predictability than the rapidly decaying VPD-only modes. Still, at shorter lead times, and particularly in summer and early fall, greater skill from the VPD-only modes means it is important to include information from both components of the system.

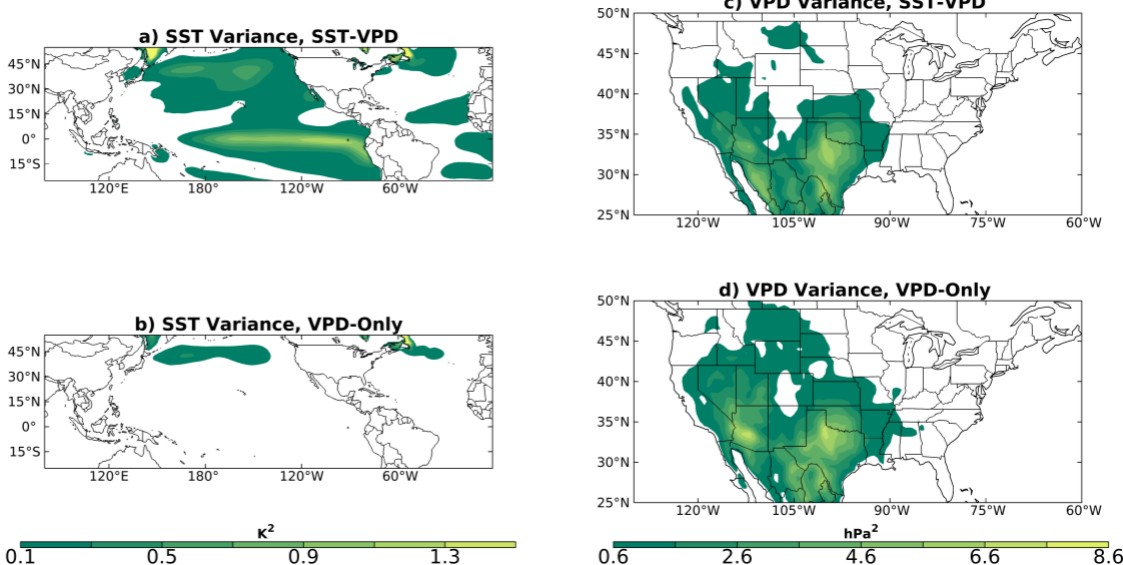

**Figure 7: Filtered SST and VPD variance for a), c) the SST-VPD and b), d) the VPD-only subspaces of the detrended LIM.**



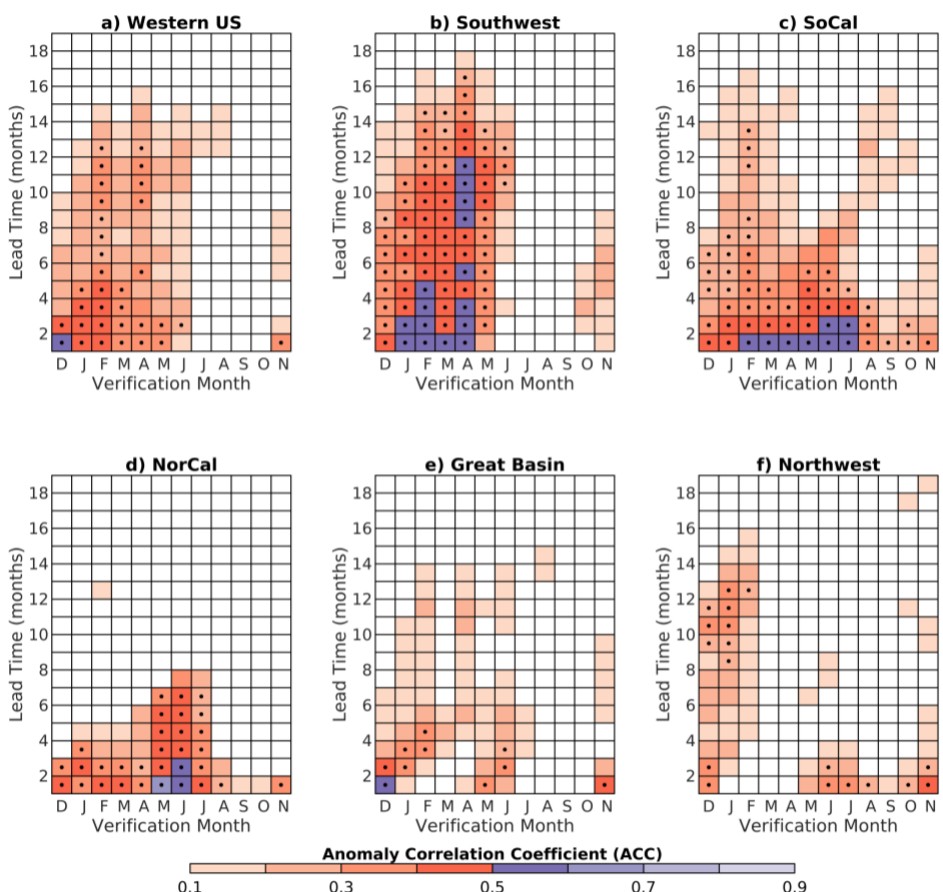

**Figure 8: as in Fig. 6 but for SST-VPD filtered ICs.**





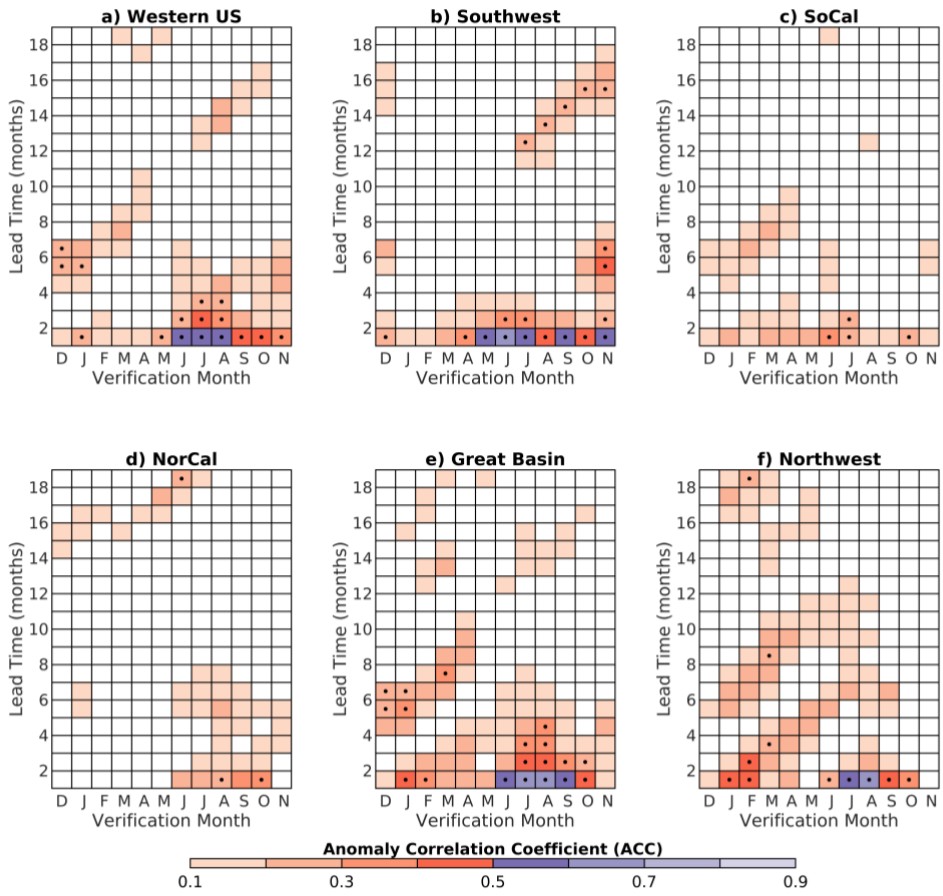

**Figure 9: as in Fig. 6 but for VPD-only filtered ICs.**

### 3.3 VPD SFOs

Periods of anomalously high skill and confidence, so-called 'forecasts of opportunity', are often considered on subseasonal-to-seasonal timescales (Mariotti et al. 2020) to understand fluctuations in forecast skill that are related to variability, as opposed to a trend. With SSTs contributing substantially to detrended VPD predictability, we hypothesize that VPD SFOs are associated with ENSO-like SST anomalies, as has been found to be the case on subseasonal timescales for a range of target variables (Albers and Newman 2021; Breeden et al. 2022a; Breeden et al. 2022b) and for reference evapotranspiration

($ET_o$) one season ahead (McEvoy et al. 2015).

Using the detrended LIM signal-to-noise ratio (Eq. 11-12), SFOs can be identified for VPD averaged over the combined Southern California and Southwest subregions for lead times up to eight months (Figs. 10-11). Three-month SFOs are indeed more skillful than non-SFOs in the Southwest and Southern California subregions, remaining true for areas farther north,

west, and south (Fig. 10a,b). Skill improvements during these events are greatest in southern California and Nevada and




western Texas, where ACC exceeds 0.5. Most of the skill during SFOs is associated with the SST-VPD eigenmodes, with the VPD-only ICs contributing to skill over the interior western US including areas in the Southwest and Great Basin (Fig. 10c,d). SFOs calculated for lead times longer than five months remain skillful but are associated only with SST-VPD modes, though recall that quadratic values such as ACC do not sum to unity (Fig. 11; Fig. S7). Three-month SFOs are initialized

most often during the JJA - DJF seasons (Fig. S8), corresponding to verifications during SON - MAM seasons. This timing aligns with the mature phase of ENSO events and when ACC peaks in these regions (Fig. 6b,c). A minimum in SFO frequency is observed for FMA and MAM initializations, the time of year known for when ENSO is often waning or changing phase.

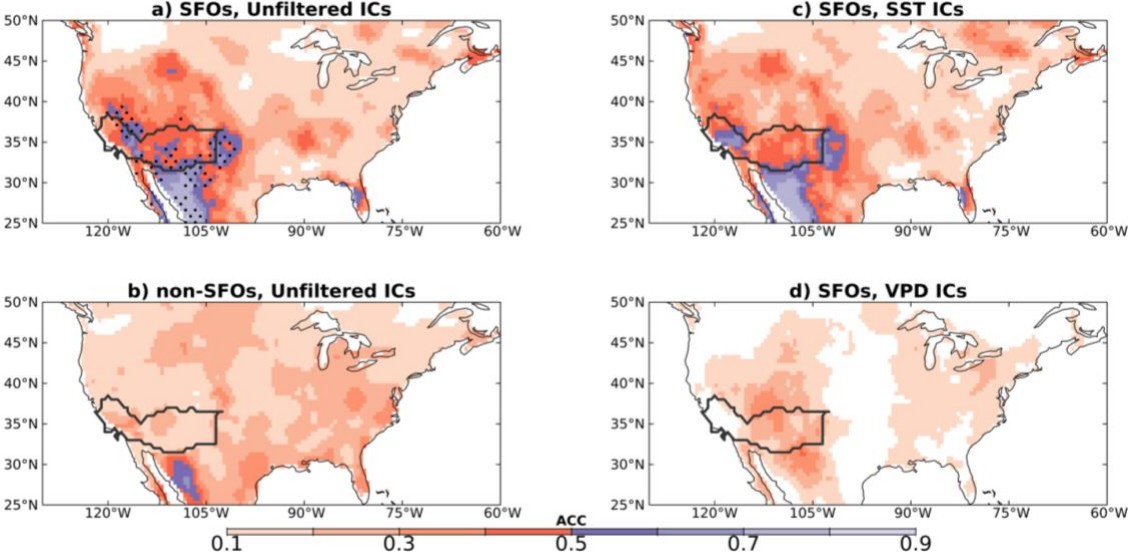

**Figure 10: Panel a) shows ACC for SFOs at a three-month lead time for the Southwest California and Southwest subregions, compared to b) the ACC for non-SFOs. Black stippling in panel a) indicates where SFO ACC is different from non-SFO ACC with 95% confidence. Panels c) and d) show the skill contributions from the SST-VPD and VPD-only subspaces for the SFOs.**

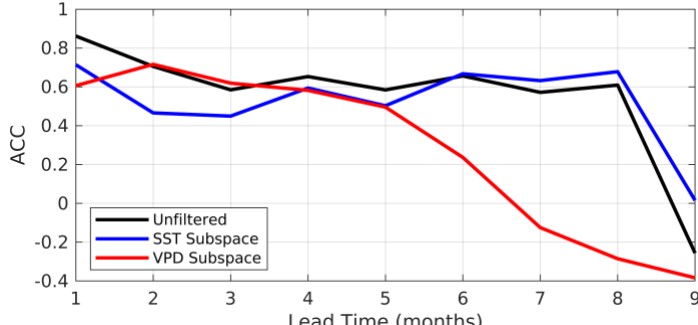

**Figure 11: ACC during SFOs, as a function of lead time, averaged over the Southern California and Southwest subregions. ACCs associated with the unfiltered ICs (black), SST-VPD ICs (blue), and VPD-only ICs (red) subspaces for SFOs are shown.**





SFOs are initialized during periods when an ENSO event is underway or expected to develop. This is shown by considering the two leading 'optimal patterns' (OPs) that maximize SST and VPD anomaly growth and include ENSO-like SSTs (OP1, 340  OP2; Section 2.2.4; Penland and Sardeshmukh 1995). Each OP has a unique set of optimal ICs (OPT-ICs) that evolve into a final pattern (OP1 or OP2) over a specified time interval. Growth varies as a function of the time interval and peaks over a seven-month interval for OP1 and a three-month interval for OP2 (Fig. S9). OP1 shows OPT-ICs resembling the Pacific Meridional Mode (Chiang and Vimont, 2004) that evolve into a mature ENSO event with SST anomalies that are strongest in the central tropical Pacific (Fig. 12). La Niña-like SSTs co-evolve with positive VPD anomalies across the southeastern 345  US and northern Mexico (Fig. 12), also the location of highest ACC during SFOs (Fig. 10). Cold eastern Pacific SSTs in OP2 co-evolve with a couplet of positive and negative VPD anomalies across the western and central US, respectively (Fig. 13). The linearity of OPs means that opposite-signed SST anomalies co-evolve with opposite-signed VPD anomalies.

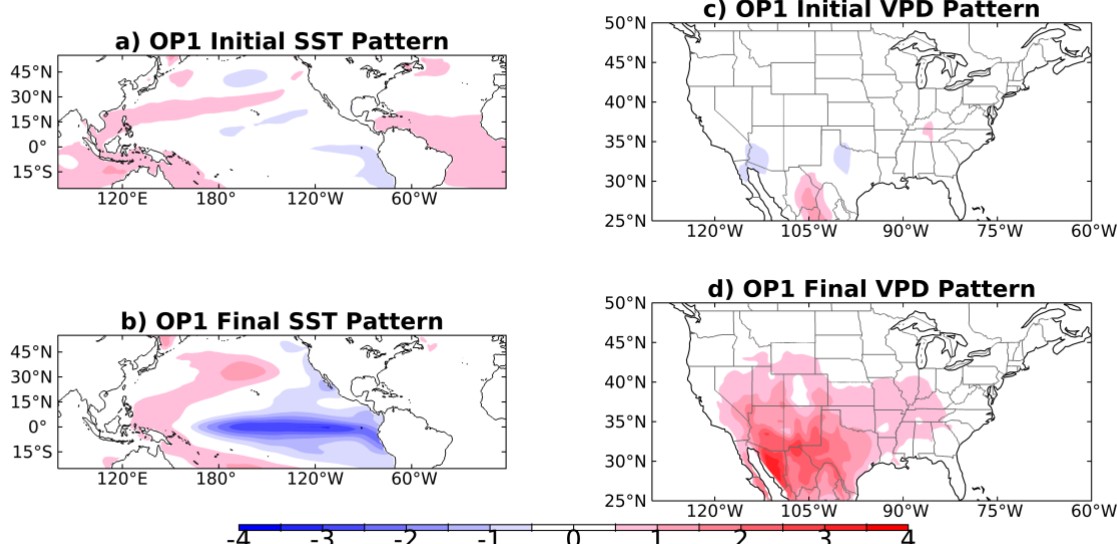

**Figure 12: Initial optimal ICs a), c), and final evolved anomalies b), d), associated with the leading optimal pattern**
**(OP1) maximizing system growth over a 7-month growth period for a)-b) SST and c)-d) VPD. Patterns are**
**dimensionless.**





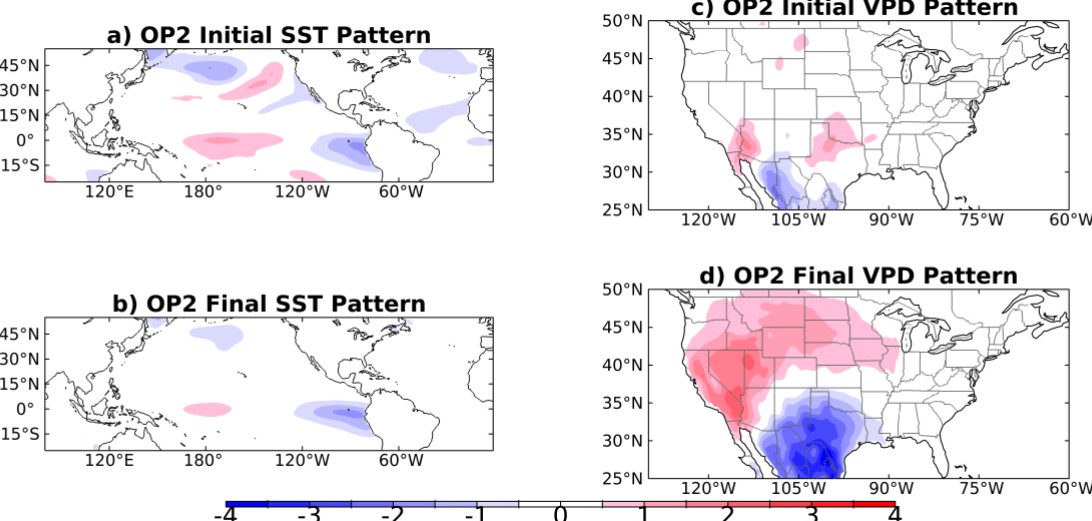

**Figure 13: As in Figure 12 but for the second leading optimal pattern (OP2) maximizing system growth over a three-month growth period.**

Comparing the ICs of SFOs and non-SFOs and their resemblance to the OPT-ICs reveals a stronger resemblance during SFOs. The amplitude of the projection of the observed ICs onto the OPT-ICs measures the similarity between conditions that trigger these patterns and each initialization, indicating the similarity between actual ICs and OPT-ICs is greater during SFOs than non-SFOs (Fig. 14). The differences in the timescale of the two OPs corresponds to differences in the lead times for which they are related to SFOs. For three-month SFOs, both the projections of OP1 and OP2 OPT-ICs display stronger projections than those observed during non-SFOs (Fig. 14a,b), while shifts during six-month SFOs are evident for OP1 but not OP2 (Fig.14 c,d). As such, the relevant patterns to SFOs depend on the lead time of interest due to the different growth periods of OP1 and OP2.




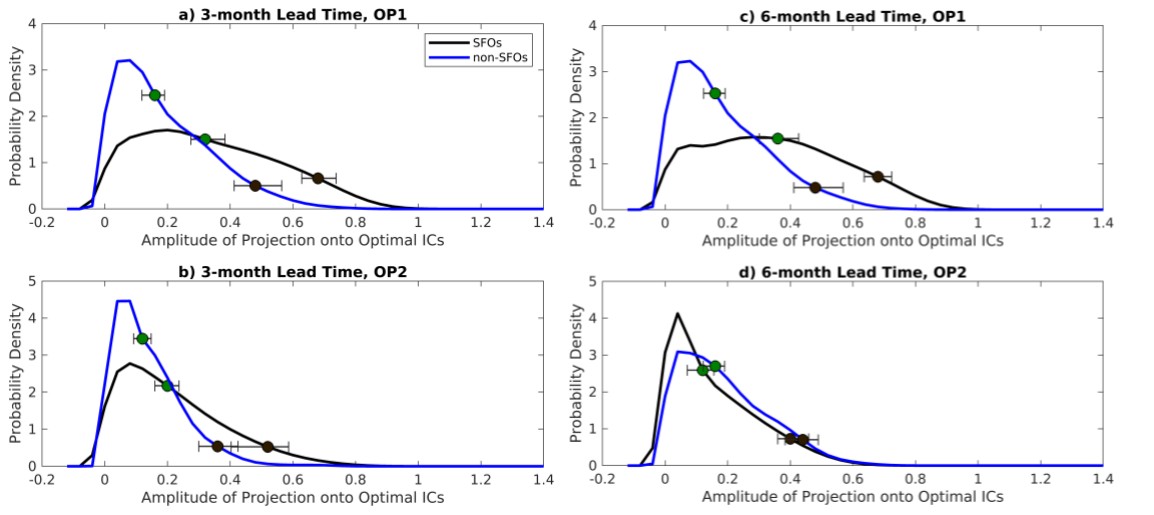

**Figure 14: PDFs of the projection onto the optimal ICs (OPT-ICs) during SFOs and non-SFOs for lead times of a)-b) three and c)-d) six months, for the two leading optimal patterns, OP1 and OP2. PDFs are shown for SFOs (black lines) and non-SFOs (blue lines) for the detrended LIM. The green and black dots indicate the median and 95th percentiles, respectively, with 95% confidence intervals denoted with error bars determined nonparametrically using bootstrapping with replacement 10000 times.**

## 4 Conclusions

This analysis demonstrates the seasonal predictability of western US VPD driven mainly by a nonlinear, long-term warming trend and seasonal-to-annual SST variability, which can be used to inform seasonal outlooks of wildland fire danger. Understanding sources of skill is important for directing efforts to improve forecast skill and anticipating when and why skill may be high or low. While skill from the trend is not episodic, it is a large component of VPD (e.g., Abatzoglou and Williams 2016); indeed, Zhuang et al. 2021 found that anthropogenic warming accounted for two thirds of the observed VPD trend amplitude from 1979-2020, eclipsing natural variability. There is seasonality in the trend contribution to skill (Fig. 5), meaning warm-season VPD – and VPD skill – is more heavily influenced by the trend than other seasons when non-trend variability, such as that related to ENSO, can obscure effects of the trend (Seager et al. 2015). Detrended VPD skill is observed from a broad verification period from December – May and is driven mainly by SST anomalies that include ENSO variability. Wildland fire danger is emerging as a year-round concern in much of the west (Westerling et al. 2006; Colorado Division of Fire Prevention and Control 2023), and our results indicate that different sources of VPD predictability should be considered at different times of year. Prediction tools should similarly be designed to capture the aggregate effect of these processes.

The timing of SST-related VPD skill in the detrended LIM corresponds to when ENSO matures, peaks, and wanes in winter and spring, consistent with the seasons highlighted in past research (Seager et al. 2015). There is minimal detrended VPD



skill in forecasts for ASO, however, meaning SSTs do not produce skillful VPD forecasts in the LIM at this time of year. Revisiting this result with a more sophisticated LIM or an alternative modeling approach is warranted, although using a seasonally varying 'cyclostationary' LIM did not improve detrended VPD skill in these months (not shown). A minimum in SST-related VPD skill during late summer and early fall is similar to the skill of reference evapotranspiration ($ET_o$), a measure of evaporative demand that is used in drought indicators and that includes VPD, in an operational forecast model (McEvoy et al. 2015). It is also consistent with what Quan et al. (2006) found using a regression model between tropical SSTs and 2mT over the continental US, which showed that the regression model had lowest skill in summer. While neither of these studies examined VPD, there may be consistencies since $ET_o$, is partially a function of VPD and therefore temperature. Our result of more frequent VPD SFOs during ENSO events also supports the findings of McEvoy et al. (2015) for $ET_o$ over the western US, suggesting VPD makes a positive contribution to $ET_o$ skill during these events.

This study does not consider the wind-driven component of fire spread, which is a critical component for a comprehensive view of wildland fire predictability. It is unclear, however, whether near-surface wind speed is predictable on seasonal timescales (McEvoy et al. 2015). Future work could also assess the seasonal predictability of metrics that incorporate many predictors including wind speed, fuels and vegetation information, such as those within the United States National Fire Danger Rating System: Burning Index, Energy Release Component, and Spread Component (Bradshaw et al. 1983; Cohen and Deeming 1985). Whether these indicators can be modeled as linear and forced by white noise, as was done for LIM-based VPD forecasts herein, remains to be determined.

**Data availability**: The JRA-55 data used in this analysis is available from the NCAR Research Data Archive: https://rda.ucar.edu/datasets/d628000/.

**Author contribution**: MLB acquired and processed data, produced figures, and wrote the manuscript text. AH, RPW, DJV and MTH contributed feedback about figures and provided edits and comments on the manuscript text. JRA shared code and provided edits and comments on the manuscript text. MRM produced a figure and provided edits and comments to the manuscript text.

**Competing interests**: The authors declare that they have no conflict of interest.

**Acknowledgements**: The authors would like to thank Matthew Newman and Sang-Ik Shin for productive conversations about linear inverse modeling.

**Financial support**: This work was funded by NOAA grant NA23OAR4050186I.

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
