# Peer review of "Seasonal Predictability of Vapor Pressure Deficit in the western United States"

_EGUsphere, 2025_

## Author Comment (AC1)

To all reviewers: We greatly appreciate the comments and suggestions for our manuscript, which have been addressed below. We also improved the colors used in the figures to be more visually appealing. The content of the figures remains the same.

Reviewer 1 Comments

General comments:

By using a linear inverse modeling framework for the fields Vapor Pressure Deficit (VPD), Sea Surface Temperature (SST), and Soil Moisture (SM), the authors forecast seasonal VPD for the western United States. The study identifies Seasonal Forecasts of Opportunity (SFOs) and links them to ENSO. The manuscript is well-written, with no significant flaws in the logic. The figures are clearly presented, and the results support the conclusions. I recommend publication of this manuscript with minor revisions. I would like the authors to consider the following comments.

We appreciate the reviewer's comments and suggestions and the time taken to review our manuscript. We have integrated the comments below into the manuscript and believe it is much improved as a result.

Major comments:

The use of soil moisture (SM) needs more justification. What would the results look like if SM were removed from the analysis? If there is a significant difference between the forecasts with and without SM, what exactly is SM capturing?

We agree that SM has a secondary impact on VPD skill compared to the inclusion of SST. Still, in the LIM including the trend, the inclusion of SM does increase VPD skill in the western US. This can be seen by considering the VPD skill in a LIM with the SM removed, indicating that the peak ACC values and coverage of skill is reduced compared to the LIM including SM (Figs R1-R2). Note that Fig. R1c and Fig. 2c are the same, because the persistence forecast does not change. We have included this information in the text and added Fig. R2 to the Supplement.

This section within the body of the text describes our hypothesis that soil moisture is capturing an aggregate and lagged effect that may influence the VPD predictability,

"Finally, soil moisture, a slowly-evolving quantity that is the aggregate effect of precipitation, vegetation, and surface fluxes on the moisture content of the soil, may also be a source of VPD predictability given its high temporal autocorrelation at depth (Rahman et al. 2015) and the potential 'reemergence' of soil moisture anomalies months after they first develop (Kumar et al. 2019)."

[Figure]

Figure R1: a) VPD ACC of a 6-month forecast from DJF for the following JJA generated by a LIM including only SST and VPD (domains and EOFs as in Table 1). b) shows the ACC of the area-averaged VPD forecasts as a function of lead time and verification month, and c) shows the same but for a full-field persistence forecast (same as in Fig. 2c).

[Figure]

Figure R2: ACC difference between the LIM including SM subtracted from the LIM without SM (SM – no SM). Positive (blue) values indicate skill improvements with the inclusion of SM.

The discussion on ENSO using the leading optimal patterns was succinct, but would it be more valuable for a general audience to include a direct analysis linking SFOs to ENSO indices? For example, similar to Breeden et al. (2022) (their Figs. 6 & 7), authors could overlay the SFOs on the Niño 3.4 index or the Oceanic Niño Index.

We have computed the relative risk of 3-month lead SFOs as a function of the Niño3.4 index and plotted the overlap of VPD SFOs with Niño3.4:

[Figure]

Figure R3: Relative risk of 3-month lead time VPD SFOs as a function of Niño3.4. The risk is relative to the unconditional probability of an SFO occurring, which is 0.15.

[Figure]

Figure R4: Three-month lead time SFOs (black dots) overlaid with the Niño3.4 index at the time of initialization.

We find that these figures provide useful context for the relationship between VPD SFOs and ENSO, with SFOs clearly increasing in frequency during strong and leading up to strong ENSO events. Consequently, Figure R4 has been added to the main results section and Fig. R3 is now included in the supplement and discussion added to the main text.

Specific comments:

L68: Beverley et al (2022) is missing from the reference list. We have included the proper reference.

L311: Mariotti et al (2020) is missing from the reference list. We have included the proper reference.

L399-401: The first two sentences about winds in this paragraph seem out of place, especially since the next sentence mentions other variables, including winds. Consider revising for clarity and consistency to ensure a smoother flow of ideas.

We have revised the text to enhance clarity.

References:

Breeden, M.L., Albers, J.R. and Hoell A.: Subseasonal precipitation forecasts of opportunity over southwest Asia. Weather and Climate Dynamics, 3, 1183–1197, https://doi.org/10.5194/wcd-3-1183-2022, 2022.

**Reviewer 2 Comments and Response**

This study explores the seasonal predictability of vapor pressure deficit (VPD) in the western United States, a region increasingly impacted by wildfires. The use of Linear Inverse Models (LIMs) to forecast VPD and decompose contributions from a nonlinear trend, SST-VPD coupling, and internal VPD variability is both technically rigorous and conceptually insightful. The manuscript is well-organized, clearly written, and the inferences drawn from its results make generally good sense.

I recommend publication pending moderate revisions. Below are several specific comments for clarification and improvement:

Thank you, we appreciate the comments and suggestions below and the time taken to review our manuscript and believe the analysis is greatly improved as a result.

1.  The authors use JRA-55 SST as input for the LIM, rather than more widely used SST datasets such as HadISST or ERSST, which are commonly validated in ENSO and SST variability studies. Given the important role of SST in the decomposition of VPD predictability, it would strengthen the manuscript to either justify this choice or test the sensitivity of results to other SST products.

JRA55 SST was selected to be physically consistent with the rest of the variables in the LIM. To verify that the results are not dependent on the SST dataset employed, we trained a new LIM using the ERSSTv5 dataset (Huang et al. 2017), which resulted in very similar VPD skill as the LIM trained using JRA55 SST (Fig R5):

[Figure]

Fig. R5: As in Figure 2 but using a LIM trained with ERSSTv5 instead of JRA55 SST anomalies. Panel c) is identical in both figures.

Since EOF truncation of SST is used to incorporate variables in the LIM, it is perhaps not surprising that a different SST datasets yields nearly similar VPD skill, since the correlation between the leading 3 PCs of JRA55 SST and ERSSTv5 are: 0.92, 0.92, and 0.95 for PC1, PC2 and PC3, respectively. This reflects the similarity in the large-scale SST covariance between the two datasets, and therefore, the similarity in the two LIMs.

We have included a statement about the lack of sensitivity to the SST field in the text and added Fig. R5 to the supplemental material.

ERSSTv5 Reference:

Huang, B., Peter W. Thorne, Viva F. Banzon, Tim Boyer, Gennady Chepurin, Jay H. Lawrimore, Matthew J. Menne, Thomas M. Smith, Russell S. Vose, and Huai-Min Zhang (2017): NOAA Extended Reconstructed Sea Surface Temperature (ERSST), Version 5.

NOAA National Centers for Environmental Information. doi:10.7289/V5T72FNM. Obtain at NOAA/ESRL/PSD at their website https://www.esrl.noaa.gov/psd/ [February 16 2025].

2. The authors calculate monthly VPD using monthly mean temperature and RH, which may underestimate extremes due to the nonlinear relationship between temperature and vapor pressure. While this method may be appropriate for seasonal-scale prediction, I think an explanation is needed for this choice over computing the monthly average of daily VPD, which is commonly used in fire-weather studies.

We have added an explanation to the text for the choice to use monthly mean temperature and RH to calculate VPD.

3. While anomaly correlation coefficient (ACC) is a standard metric, it does not capture magnitude errors or bias. For example, what are the root mean square error (RMSE) values corresponding to Figures 1 and 2? Including such metrics would provide a more comprehensive assessment of model performance.

We have now calculated the RMSE corresponding to the first two ACC figures (Figs. 2-3), confirming that forecasts with statistically significant skill are also characterized by RMSE values less than one standard deviation of VPD (now Figs. S8-S9 in modified supplement and shown below).  We have added discussion of RMSE to the main text.

[Figure]

Figure R6: As in Figure 2 but for root-mean-square error (RMSE) normalized by the standard deviation of VPD anomalies.

[Figure]

Figure R7: As in Figure 3 but for RMSE normalized by the standard deviation of VPD anomalies.

4.  It is not quite clear how the SST pattern in Figure 12a resembles the Pacific Meridional Mode (PMM). Could the authors consider quantifying this similarity, for example by providing a spatial correlation coefficient with a canonical PMM pattern?

Upon further inspection, the PMM pattern is characterized by opposite-signed anomalies in the subtropical east Pacific and tropical east Pacific, which differs from the OP1 OPT-ICs. We have removed mention of PMM from the text.

5.  Could the limited detrended VPD skill in ASO be attributed to the weak influence of SST on large-scale atmospheric circulation over the US during this season? The authors might consider discussing other factors (e.g., Northern Hemisphere teleconnection or Arctic sea ice) here, which may also play a more important role.

We have reviewed the publications below and added mention of alternative sources of seasonal VPD predictability that are not considered in the present analysis to the Conclusions and Discussion section.

References:

Zou, Y., Rasch, P. J., Wang, H., Xie, Z., & Zhang, R. (2021). Increasing large wildfires over the western United States linked to diminishing sea ice in the Arctic. Nature communications, 12(1), 6048.

Liu, S., Hu, S., & Seager, R. (2024). The West Pacific teleconnection drives the interannual variability of autumn wildfire weather in the western United States after 2000. Earth's Future, 12(11), e2024EF004922.

Lou, J., Joh, Y., Delworth, T. L., & Jia, L. (2025). Identifying source of predictability for vapor pressure deficit variability in the southwestern United States. npj Climate and Atmospheric Science, 8(1), 139.

---

## Author Response (AR2)

Response to Reviewer Comments on "Seasonal Predictability of Vapor Pressure Deficit in the western United States"

8/22/25

**Co-editor decision: Publish subject to minor revisions (review by editor)**

by Dariusz Baranowski

Public justification (visible to the public if the article is accepted and published):

The two reviewer's reports are in and both agree that the authors well revised the manuscript and addressed all comments and concerns raised by them.

The manuscript can be accepted, subject to a minor revision, following the report from the reviewers. Specifically, address one comment and one suggestion:

1. L440. Replace 'western US wildfires' with 'autumn western US wildfires' for greater accuracy.

We have added 'autumn' in the suggested location.

2. I think some readers may be curious about the dynamic mechanisms in Figures 13 and 14 that could explain the difference between Figure 13d and Figure 14d. While this is beyond the scope of the paper, it would be beneficial if the authors could briefly highlight it as a direction for future work in the discussion. For example, how do different SST patterns influence VPD in the western United States?

We have added this promising avenue of future work to the discussion and conclusions section.

That's it.

Thank you, we greatly appreciate the time taken to review our manuscript!